# A Process Evaluation of the UK Randomised Trial Evaluating ‘iSupport’, an Online e-Health Intervention for Adult Carers of People Living with Dementia

**DOI:** 10.3390/bs15081107

**Published:** 2025-08-15

**Authors:** Patricia Masterson-Algar, Fatene Abakar Ismail, Bethany Anthony, Maria Caulfield, John Connaghan, Kodchawan Doungsong, Kieren Egan, Greg Flynn, Nia Goulden, Zoe Hoare, Gwenllian Hughes, Ryan Innes, Kiara Jackson, Suman Kurana, Danielle Proctor, Rhiannon Tudor Edwards, Aimee Spector, Joshua Stott, Gill Windle

**Affiliations:** 1School of Health Sciences, Bangor University, Bangor LL57 2EF, UK; 2Department of Computer and Information Science, University of Strathclyde, Glasgow G11 XH, UK; 3Centre for Health Economics and Medicines Evaluation (CHEME), School of Health Sciences, Bangor University, Bangor LL57 2EF, UK; 4Centre for Applied Dementia Studies, Faculty of Health Studies, University of Bradford, Bradford BD7 1DP, UK; 5North Wales Medical School, Bangor University, Bangor LL57 2EF, UK; 6ADAPTlab, Department of Clinical, Educational and Health Psychology, University College London, London WC1E 6BT, UK

**Keywords:** process evaluation, dementia, e-health, intervention, unpaid carers, RCT, psychoeducation, complex interventions, mixed-methods

## Abstract

Supporting dementia carers is a global priority. As a Randomised Controlled Trial (RCT) (n = 352) of the Word Health Organization recommended, an internationally disseminated ‘iSupport’ e-health intervention was conducted, revealing no measurable benefits to the wellbeing of adult dementia carers. This process evaluation contributes original insights of the trial outcomes. Its aims were to ascertain the usability and acceptability of iSupport, participant engagement and adherence to iSupport, and contextual factors influencing its implementation and potential impact. The process evaluation followed a mixed-method design. The following data were collected from all participants randomised to iSupport (n = 175): (1) post-intervention evaluation questionnaire (n = 93) containing the 10-item System Usability Scale and bespoke items exploring acceptability, engagement, and perceived impact; (2) qualitative interviews (n = 52) with a sub-sample of participants who were purposively sampled according to age, scores on the outcome measures, and gender, as these interviews aimed to generate contextual detail and explanatory accounts; and (3) ‘Access’ data from the iSupport platform (n = 175). Descriptive statistics was used to report on the frequency of survey responses whilst a thematic analysis approach was followed to identify themes from the qualitative interview data. Data sets were analysed independently and then used with respect to one another in order to generate explanatory pathways related to the usability, acceptability, and the impact of iSupport. Despite good trial retention, 8.3% of participants (n = 32) did not spend any time on iSupport, and 54% (n = 94) spent between 30 min and 1.5 h. Factors driving this were the following: time constrains, method of delivery, and content characteristics. Positive impacts of iSupport were also described. Participants, including those with extensive caring experience, reported how iSupport had made them feel reassured, valued, and more able to ask for help. They also reported having an improved outlook on their caring role and on the needs and feelings of the person living with dementia. Research and practice should focus on exploring blended delivery, including self-directed and interactive components, such as regular contact with a health professional. These insights are critical for supporting the global implementation and adaptation of iSupport and offer valuable directions for future research.

## 1. Introduction

Across the world, family and friends provide significant levels of care for people living with dementia ([30]). Known as ‘unpaid carers,’ they provide substantial and complex care, which often intensifies over time due to the degenerative nature of dementia ([25]). Although there are positive aspects to caring, such as personal growth, strengthened relationships, and a sense of purpose ([50]; [56]; [38]), there is a wealth of evidence showing how caring is associated with reduced wellbeing and ill-health including stress and depression ([21]; [20]; [42]). This often linked to heavy caregiver burden ([13]) and social isolation ([24]).

Considering the estimated 50 million people affected by dementia worldwide ([1]), national and international policies are advocating for the provision of adequate training and support for unpaid carers. Hence, in recent years much of the research efforts have focused on developing and testing support interventions for this population ([18]; [6]). Specifically, and in part spurred by the COVID-19 pandemic, self-guided e-health interventions have been proposed as an alternative to face-to-face options, offering a supportive environment and mitigating some of the negative impacts on caring ([2]; [17]; [37]; [31]). These interventions have the potential to reach large numbers of carers regardless of their geographical locations, at a low-cost and at a place and time convenient to them ([44]; [19]; [26]).

‘iSupport for dementia carers’, originally developed by the World Health Organization (WHO), is a self-guided online training programme based on principles of psychoeducation ([43]). It is currently being adapted in over 40 countries around the world (e.g., [47]; [52]; [54]; [35]). However, we are the first to evaluate clinical and cost effectiveness of iSupport in an English-speaking population via a randomized controlled trial (RCT). This is the largest RCT of iSupport in any country to date ([51]) and was registered with the ISRCTN on the 9 March 2021 (TRN 17420703). While the RCT was able to reach recruitment and retention targets, trial results showed no measurable benefit in any of the primary (depression and distress) or secondary (resilience, dementia knowledge, quality of the care relationship, and anxiety) outcome measures ([51]).

As recommended by the Medical Research Council (MRC) UK guidelines for complex interventions ([36]), a process evaluation was conducted alongside the iSupport RCT. Process evaluations serve a vital role in providing a framework to identify the underpinning characteristics of the overlapping intervention components and how these factors contribute to the delivery, acceptance, and implementation of the complex intervention to a set standard ([46]; [39]; [36]). More specifically, process evaluations should also aim to provide detailed insight into how trial participants experience their engagement with a complex intervention (such as iSupport) ([36]). By embedding a process evaluation, trials such as the iSupport RCT will produce higher quality and context-sensitive results which in turn can be instrumental in generating theory-informed recommendations to guide future research and implementation efforts in practice ([8]).

In this paper we report on the results of the process evaluation of the iSupport RCT. The overall aim was to evaluate and understand what worked (or did not) and to provide key insights into reasons for trial results that can support the clinical implementation of findings and identify future research directions. This process evaluation was also vital at identifying the underpinning characteristics of iSupport and contextual mechanisms of impact. Specific objectives were the following:

To examine the usability and the acceptability of iSupport.

To determine participant level of engagement and adherence to iSupport.

To examine the contextual factors influencing the uptake and implementation of iSupport.

To examine the extent to which iSupport may have changed behaviours beyond the intervention (e.g., help-seeking).

## 2. Materials and Methods

iSupport is a psychoeducation e-health intervention that consists of 23 lessons distributed across five modules: introduction to dementia; being a carer; caring for me; providing everyday care; and dealing with behaviour changes ([43]). Each lesson comprises relevant information on the topic, caregiving scenarios, and interactive skill training exercises. Participants in the intervention arm of the iSupport RCT were given access to iSupport which they could access for six months through a personal computer, tablet, or mobile phone. Although no ‘dose’ was specified, participants were advised to use it ‘regularly’ to obtain the most benefit. The RCT built-in the provision of an ‘e-Coach’, a member of staff who contacted participants (via email) at three time points to offer technical support (shortly after randomisation, 1-month post-randomisation, and once more for participants who had not used the intervention by 2 months).

### 2.1. Design of the Process Evaluation

This mixed-methods, theory-informed process evaluation followed the MRC guidance for process evaluations ([36]). A logic model (Appendix A), describing underlying assumptions, logical linkages, inputs, and short-, medium-, and long-term outcomes, was created informed by [23] ([23]) and [33] ([33]). We were further informed by the Consolidated Framework of Implementation Research (CFIR) ([15], [16]), Normalization Process Theory (NPT) ([32]), and [45] ([45])’s acceptability framework, all highly attuned to the challenges of complex interventions. These theories guided the process evaluation parameters ([40]) to explore processes and interactions taking place between the iSupport intervention and the changing and diverse context. As recommended by [22] ([22]) Figure 1 presents a diagram representation of the research process showing the methods used and, vitally, how they are connected (points of interface/integration).

The iSupport RCT and the process evaluation were granted ethical approval on 12 April 2021 by Bangor University’s School of Medical and Health Sciences Academic Ethics Committee (AEC), reference number 2021-16915.

### 2.2. Participants and Data Collection

To address the aims and objectives we used a mixed-method approach to data collection focusing on all the participants in the iSupport group (n = 175). The following data collection methods were used:

*Post-intervention evaluation questionnaire:* A link to a tailor-made online survey (via OnlineSurveys.com) (Appendix A) was sent to all intervention participants who completed their six-month follow-up (T2; n = 126). The online survey was self-completed and consisted of the following:▪A bespoke survey including Likert scale and open-ended-type questions: In order to explore participants’ experiences of caring for a person with dementia and engaging with iSupport, we developed questions addressing the five constructs in the NPT, as recommended by [32] ([32]).▪The System Usability Scale (SUS, [11]): This 10-item scale evaluates the overall usability of the ‘iSupport’ platform. Each item is a statement (e.g., “I thought ‘iSupport’ was easy to use”), and responses were given on a 5-point Likert scale from 0 (strongly agree) to 4 (strongly disagree). Scores between 0 and 50 indicate a ‘not acceptable’ tool and 50 and 70, a ‘marginally acceptable’ tool, and finally a score of 70 and 100 indicates the tool is ‘acceptable’ ([4]).

*iSupport usage data:* The frequency and length of use (time spent on iSupport and number of ‘log ons’) was collected for all intervention participants (n = 175). This pseudo-anonymised data was securely extracted from the intervention platform hosted by the Pan American Health Organization every two weeks and later collated, for each participant, at the end of their six-month intervention period.

*Access and use of the e-Coach (technical support):* For all intervention participants the frequency and content of contacts with the e-Coach were securely recorded in an Excel spreadsheet.

*Semi-structured qualitative interviews:* These were conducted via videoconferencing software (e.g., Teams and Zoom) or by telephone with a sub-sample of participants (target n = 50) in the intervention group following their six-month follow-up in the trial. A purposive sampling matrix was created which included age and gender, along with baseline scores of the two primary outcomes of the RCT, the Zarit Burden Interview (ZBI-12; [5]), and the Centre for Epidemiological Studies of Depression Scale (CES-D-10; [3]). This pragmatic approach was chosen to hear from participants in different groups and with different experiences. Interviews were conducted by the trial team’s research assistants trained in using the interview schedule and qualitative interviewing. The interview schedule (please see Appendix A) was developed following the CFIR constructs. Additionally, in order to ensure an in-depth exploration of issues around acceptability of iSupport we were also guided by the seven constructs proposed by [45] ([45]) in their theoretical framework for acceptability. Written consent from all participants was obtained prior to interview. Reasons for declining participation were also noted to understand any barriers to participation and potential selection bias. All interviews were audio recorded, professionally transcribed verbatim, and anonymised.

All personal information were handled in line with the UK Data Protection Act (2018), the General Data Protection Regulation (EU GDPR) 2016/679, and local university policies.

### 2.3. Data Analysis

Both qualitative and quantitative data sets were used to describe the multiple, multi-level interacting components of the iSupport intervention. Trial process evaluation data was analysed independently of the outcome evaluation data (results published elsewhere, [51]).

Descriptive statistics were used to analyse the online survey responses and iSupport platform data. Total numbers, percentages, means and standard deviations, or the median and range, if not normally distributed, were calculated.

Qualitative data (interview transcripts) was analysed using the [10] ([10]) thematic analysis approach (see Figure 1). An initial coding framework was developed (Appendix A) reflecting all aspects of the intervention and its implementation and informed by the constructs in the CFIR ([15]) and in the acceptability framework proposed by [45] ([45]). A thematic analysis ([10]) was then carried out in two phases (see Figure 1). During Phase 1, two researchers (PM-A and MC) coded the transcripts of the first 26 interviews that were completed and assigned relevant data extracts to each code. The distribution of codes was recorded, and a small number of new codes were created for data falling outside the coding framework to avoid missing important concepts. No codes were removed. Emerging preliminary themes were identified as meaningful patterns across coded data, and as a result an initial programme theory and a refined coding framework were developed. Phase 2 involved the thematic analysis of the remaining transcripts (n = 26) applying the refined coding framework. An iterative process of review and discussion among process evaluation research team members was followed to agree on a set of preliminary themes (Appendix A).

#### Synthesis

As shown in Figure 1, the synthesis across all data sources took place once data sets were analysed separately. As [22] ([22]) recommends, our mixed-method design was guided by two descriptive dimensions: Firstly, the ‘timing of the integration between our data sets’, in which interview data and quantitative survey and platform data sets were collected and analysed independently, and then used with respect to one another, and secondly, the ‘purpose of the integration’, in which qualitative interview data were used to identify themes and quantitative data were then mapped onto these themes to see whether it contradicted or supported them. To complete this mapping exercise, an online workshop was organised and delivered by the first author (PM-A) and attended by members of the research team (n = 5). Prior to the workshop PM-A shared a document which included a detailed summary of the results of both, the quantitative and qualitative data analysis. As a result, explanatory pathways related to the usability, acceptability, and impact of iSupport were generated which provided an in-depth understanding of the RCT results.

## 3. Results

To reach the target sample of 50, 83 participants were invited to take part in the semi-structured interviews. A total of 13 did not reply to the invitation, and 15 did not consent, time constraints being the main reason for declining. A total of 52 interviews were completed. All participants chose to conduct the interview using an internet-based service (Teams or Zoom). The interview participants’ demographic characteristics and baseline data are summarised in Table 1. In line with overall trial participants characteristics, those who took part in interviews were 90.4% white British, and the majority were women (69.2%); 51.9% were carers for their parent, and 46.2% were carers for their spouse/partner. A total of 42 of these (80.8%) spent 30 min or more on iSupport (see Appendix A).

A total of 126 intervention group participants completed the six-month follow-up data collection point (T2), after which they were sent an evaluation questionnaire (via an OnlineSurveys.com link); 93 participants (74%) completed the survey (see Table 1 for survey participants’ demographic characteristics and baseline data). Among these participants, 78.5% (n = 73) used iSupport for 30 min or more (see Appendix A). The SUS score was calculated with data for these 73 participants. The average participant’s ratings on the usability of iSupport was of 58.77 (SD 19.75). The median was 65 [45, 72.5], suggesting a ‘marginally acceptable’ perception of the programme’s usability ([4]). Thirty-three participants, amongst these 73, contacted the e-Coach during the study with comments and/or questions. These were mainly related to difficulties logging on to iSupport or navigating the modules and lessons. Younger participants (n = 42; 18–64 yrs.) revealed a slightly less positive perception (median = 63.75; IQR 44.4–70) than participants 65+ yrs. (n = 31; median = 67.5; IQR 45–72.5). Spearman’s non-parametric test showed no significant correlations between age and SUS scores [*r*_s_ = 0.007, *p* = 0.956] and time on iSupport and SUS scores [*r*_s_ = 0.165, *p* = 0.164].

The e-Coach sent out a total of 387 emails to intervention group participants. A total of 80 intervention-related questions were resolved by email, and 2 questions resolved over the phone, from 33% (n = 58) of intervention participants. Among the participants who completed the online survey (n = 93), 68% (n = 50) answered that they did not need to contact the e-Coach, and among those who did (n = 23), 61% (n = 14) considered the help they received was effective.

### 3.1. iSupport Uptake

Data synthesis across all data sources revealed how participants did not access iSupport as expected. The time that intervention group participants (n = 175) spent on iSupport was low. The median length of time spent using iSupport was 49 min (IQR 5–104). A total of 18.3% of intervention participants (n = 32) did not spend any time on iSupport, and 54% (n = 94) spent between 30 min and 1.5 h. One participant spent more than 7 h (Figure 2). Amongst intervention participants, those that took part in the process evaluation showed a similar pattern of use with 54.8% of participants who completed the online survey and 57.7% of interview participants spending between 30 min and 1.5 h on iSupport (Figure 2).

### 3.2. Patterns and Driving Factors Behind Level of Uptake

Synthesis across all data sources revealed several factors driving the uptake and level of engagement of participants with iSupport (see Figure 3). These are described in detail below.

#### 3.2.1. Time Constrains

Participants struggled to find the time to engage with iSupport, and their level of ‘caring burden’ was the main reason. Completing iSupport was described as an ‘added task’ to an already busy schedule:


*“To me it became another chore if you like. It became another task in a day that I needed to do. And it was a task that actually I could say well, I haven’t got time to do that, or that’s not the best use of my time today”.*

*(P1, female spouse, aged 68)*



*“I am trying to juggle everything, and do this, and it’s nice to have the support there, for someone to say you have to value yourself, and you can have time out to do things, but in reality, it just feels like another pressure, and it really gets on my nerves.”*

*(P2, daughter, aged 56)*


The median length of time spent caring for participants who took part in the qualitative interviews was 3 years (IQR 1.5–5), and their median baseline ZBI score was 21 (IQR 16–26) with 60% of them scoring ZBI of >21 which indicated high levels of distress. This is in line with RCT baseline data from all intervention participants. It is likely that this high level of burden could have influenced participants decision-making in terms of how much to prioritise using iSupport.


*“I must admit that it, it came well down the bottom of, of the to-do, it wasn’t, it was never a priority, if there was something else that needed doing then I did the something else instead”.*

*(P3, female spouse, aged 63)*


Interestingly, survey data showed that out of the 73 participants who spent a minimum of 30 min on iSupport, 46.6% (n = 34) identified ‘Time needed’ as an ‘extremely important’ or ‘important’ factor underpinning their use of iSupport (see Appendix A).

The ‘pattern of use’ was identified as a factor closely linked to the level of uptake and engagement. Those participants who were able to find the time to use iSupport followed different patterns of use. Some completed it in one long session (‘in one go’) or in a few long sessions. Two participants explained:


*“I’d say it was a few longer sessions. I’d get stuck into one of the modules and work my way through it, partly because I thought, okay, I’m in it now. I know where I am. I’ll just go through”.*

*(P4, daughter, aged 67)*



*“I have very limited time…I ended up going through the modules, basically as quickly as I could, in one go. I did them over a couple of days”.*

*(P5, daughter, aged 60)*


Other participants acknowledged that due to the considerable amount of information contained in iSupport they chose to ‘dip in an out’ as and when they found time:


*“I tended to do it in sections because, you’ve got the modules and then various sessions. I’d go on for maybe an hour or two at a time and just slowly work my way through…It wasn’t a case of sitting down and just spending two solid days working through it and completing it one go. It was a case of dipping in and out to suit the sort of domestic circumstances I was faced with at that time”.*

*(P6, son, aged 66)*


Most survey participants (63%) reported having completed all five modules of iSupport. Interview participants reported following the module order; however, the relevance and resonance of the content to their personal and caregiving circumstances influenced their pattern of use and level of engagement with iSupport. When the information was relevant to them, participants reported spending more time on iSupport.


*“I did follow the module orders. Sometimes I went back over some of the modules when they were relevant to my own circumstances to read them again as I was going through”.*

*(P7, male spouse, aged 62)*


#### 3.2.2. Method of Delivery

Participants described the process of accessing iSupport and navigating through the modules and lessons as, at times, complex. Interview data revealed some of the difficulties that participants experienced in ‘navigating iSupport’, with frustration reported at not being able to ‘pick up where they left it’ easily. One participant explained the following:


*“I found it very difficult. The only way I could solve it was I had to go back to the beginning every time and then start page by page, going through… I felt it could have been simpler and needed to be simpler. I was on the verge of giving up”.*

*(P8, male spouse, aged 76)*


Frustration was compounded for participants who did not efficiently engage with iSupport, despite considering it a useful tool:


*“I thought it was a bit too cumbersome, but it’s the beginning of a useful tool, but it does need modifying and making it easier to use. It’s not difficult, but a bit cumbersome”.*

*(P9, male spouse, aged 71)*


Interestingly, online survey responses from participants showed that, when asked to rate the importance of the ease of use and complexity of the iSupport platform as a challenge to engagement, 16.4% (n = 12) rated it as ‘extremely important’ (5). However, 23 participants (31.5%) did not consider it important (1,2 rating) (see Appendix A).

Closely linked to issues around the complexity of navigating iSupport, IT literacy was identified as a further factor impacting on level of uptake of iSupport. Survey responses showed that 37% of participants (n = 27) felt confident in using technology (see Appendix A). A total of 79.4% (n = 58) reported being able to use iSupport without any help, and 93.2% (n = 68) reported having the right resources to be able to use it (see Appendix A). However, when asked their opinion on whether most carers would be able to use iSupport without help, participants’ responses were more widely spread, and 34.3% (n = 25) of them did not think they would.

Online survey data showed that 95% of participants accessed iSupport from their home using a tablet (n = 16 [21.9%]), laptop (n = 33 [45.2%]), or desktop (n = 11 [15.1%]). Among interview participants, the 100% online delivery aspect of iSupport was seen as a positive by some: *“The fact that it was online was just so much better. I would never have looked at it if it was in a booklet”. (P10, daughter, aged 55)*.

However, most participants reported a preference for a mixed approach including the option of a printed booklet alongside the web-based version. In some cases, this was linked to a lower level of IT literacy:


*“It depends on how computer-oriented they are. I’m back to wondering whether it’s an age thing. I don’t really know. I wasn’t brought up with computers”.*

*(P11, female spouse, aged 78)*



*“I did wonder whether if I had a booklet, it would be easier than going on the internet for some people. It wouldn’t be so intimidating perhaps”.*

*(P12, daughter, aged 65)*


For those confident in using IT this was due to personal preference: *“I work all day on screen, you know, being a researcher; the last thing I want to do is spend one minute longer looking online at some resource” (P13, daughter, aged 54).*

Online survey data showed that 83.6% of participants (n = 61) viewed the self-paced and self-directed nature of iSupport as an advantage. More specifically, it showed that participants (37%) valued, for example, the fact that it allows users to choose the right content for them (see Appendix A). However, interview data revealed how the limited feedback provided by iSupport (only standard answers to the ‘tick box’ exercises) played a role in influencing participants’ low level of engagement.


*“I didn’t finish all the modules because I found that when I was filling in answers, and clicking finish, there was no feedback, so one almost felt I could have put anything, and clicked finish and onto the next one”.*

*(P1, female spouse, aged 68).*


Interview participants strongly endorsed a blended approach to delivery embedded within wider support systems (e.g., peer support groups):


*“I would recommend it on the basis that it’s there as part of a package of support, with a person attached to it as well. I think if it’s not just on its own, but it would need to be something that’s done in combination with having access to sort of local support”.*

*(P14, daughter, aged 57)*


More specifically, some participants recommended the addition of an interactive face-to-face component to iSupport. They explained how they would have benefited more from iSupport if they had had the chance to discuss it with a health or social care professional, for example an Admiral Nurse (name given to specialist dementia nurses in the UK), who could potentially help them identify the aspect of iSupport more relevant to their own situation:


*“What I would hope they get is then some face-to-face or one-to-one contact with somebody like an Admiral Nurse where they can just help sift through the things that’s important for their situation”.*

*(P15, male spouse, aged 49)*


#### 3.2.3. Content Characteristics

Overall, participants valued the content of iSupport and were positive about its ‘look’ and presentation; they considered that it was clear and simple. One participant put it as the following:


*“It was uncluttered. It kept it fairly simple, nice big writing, not loads of wordy passages to work through. And the graphics were simple and clean and concise”.*

*(P16, son, aged 62)*


Some participants described iSupport as having an *‘old fashioned’* look and made suggestions for improvement: *“You probably could have had more examples, photographs. You could make it visually a bit more modern looking” (P4).* Some participants reflected on how the device used to access iSupport may have affected its usability: *“I was using it on an iPad, so maybe that didn’t help, but the writing was quite small. There were bits of space, and the colours were a bit muted at times…”. (P17, daughter in law, aged 57).*

However, interview data revealed a tension between the generalisable nature of its content and a desire for more specific and personalised information. For some participants the broad and general coverage of content was a positive attribute of iSupport:


*“It covered a round gamut of everything that was suggested. I think it was perfect… it’s a wide variety of different activities that I could either react to or not react to if it wasn’t appropriate for myself”.*

*(P18, male spouse, aged 68)*


For others, however, this resulted in the content not resonating with their circumstances, thereby leading to a decrease in their level of engagement: *“I tried to flick through to later bits, and it seemed quite generic, and then I gave up. I thought I think I’m going to be wasting my time here”. (P19, daughter, aged 54)*.

Online survey data revealed that participants held a positive opinion regarding the relevance of the modules of iSupport to their role as carers (Table 2). Of the online survey respondents who spent a minimum of 30 min on iSupport (n = 73), most considered the content of all five modules as ‘extremely relevant’.

More specifically, some participants valued the scenarios included in iSupport as a framework for self-reflection: *“It was the format and the activities that made me think about this journey” (P18, male spouse, aged 68).* But others desired content that included practical support and a reference guide with straightforward tips and advice, as one participant explained:


*“The content was getting lost in all these different scenarios and examples and situations. Instead of 190 pages with all these scenarios, cut back on some of them and try and just get across the salient facts that people should be taking on board”.*

*(P6, son, aged 66)*


#### 3.2.4. Target Audience

As previously discussed, a considerable number of participants who completed the online survey (n = 44, 50.3%) considered that the content of iSupport resonated with their daily role as carers (see Appendix A). However, interview data revealed inconsistencies with regard to the target audience. The level of caring experience (e.g., years caring) and the stage of dementia, ranging from mild to severe, were identified as crucial factors influencing the relevance of iSupport. This was particularly notable for very experienced carers for whom the content was seen as ‘too basic’:


*“Some of the things that I was reading, I’m thinking but what else? I need more than this, because when you’ve been caring for somebody for a long while, you’ve gone through all of that”.*

*(P20, daughter, aged 65)*


For those new to caring, iSupport was described as *“an entry point into getting the landscape of things” (P13, daughter, aged 54).* Interestingly, some of the content in Module 5, looking at symptoms and behaviours that are often more frequent in later stages of dementia, was seen as ‘*too much too soon*’ for some participants.


*“I sort of veer away kind of reading about it too early because I really don’t want to know what’s ahead to some degree”.*

*(P10, daughter, aged 55)*



*“I think it’s a kind of fear, a reluctance to look at. I still think it’s almost a dream, it’s all going to go away tomorrow. It’s talking about everyday care, and care homes and things like that. I was reluctant to look at those”.*

*(P17, daughter in law, aged 57)*


### 3.3. Perceived Impact—Behaviours

Evidence across all data sources revealed a varied picture regarding perceived impacts of iSupport. According to a significant number of participants who completed the survey, their anxiety, depression, or stress ‘did not improve at all’ (Table 3), which supports the findings from the RCT ([51]).

Despite this, interview data revealed several positive impacts linked to time spent on iSupport. Firstly, participants, even those with extensive caring experience, reported iSupport reassured them in how they approached their caring role. Consistent with this, survey data indicated that 45.2% of participants (n = 33) felt their confidence as a carer had improved (Table 3).


*“It’s just sometimes reassuring to read things that endorse either how you think or that you’re doing something right or that you’re allowed to feel like this. And that is a good takeaway”.*

*(P8, male spouse, aged 76)*


Secondly, participants felt more able and entitled to ask for help and think about their own wellbeing.


*“It made me appreciate that I’d got to look after myself, because you get so far into your carer’s role, that it’s all encompassing, you forget that if you’re running on empty, you’re not going to be able to care for your loved one the way you want to”.*

*(P21, daughter, aged 57)*


Thirdly, iSupport’s specific focus on carers made them feel acknowledged and valued in their role, addressing their needs specifically.


*“When I think that (iSupport) is only for carers, that’s important, to kind of acknowledge the very fact that they’re attempting to take on this role is significant”.*

*(P14, daughter, aged 57)*


Finally, participants reported feeling better informed and equipped to deal with the challenges of caring, particularly with regards to the type of support needed.


*“I felt like I had a lot more information at hand. I had more evidence maybe, to back me up in some of the conversations I was having with social workers as well. About trying to get some help and about what was important for us as carers”.*

*(P22, daughter, aged 42)*


This sentiment was also reflected in online survey data responses (see Appendix A) which overall indicated that a significant proportion of participants considered that iSupport had been a helpful tool for increasing their knowledge about dementia (19.2% strongly agree/46.6% agree) and their caring role (21.9% strongly agree/46.9% agree).

During interviews, participants reported having an improved outlook regarding not only their caring role but also the needs and feelings of the person living with dementia. This, in turn, enhanced the quality of the caregiving relationship:


*“It made me step back and think about how I speak to Mum and perhaps how some of the things I say to her make her feel, which was sometimes a difficult read but going forward, has helped and will continue to help”.*

*(P23, daughter, aged 52)*


Amongst interview participants there were also some, particularly those with years of experience of caring (Table 1), who reported no impact of iSupport. They attributed this to iSupport not offering any new or relevant information to address their specific caregiving circumstances:


*“Because of the position we were at, I was more just nodding head in agreement and saying yeah, that’s right. There wasn’t a lot I could get from it, because of the stage I was at. And I felt that it didn’t address the advanced dementia particularly well”.*

*(P24, male spouse, aged 76)*


Lack of impact was also linked to participants feeling like they did not ‘need’ iSupport at that point in time, as they were experiencing a low level of burden.


*“I think, on the whole, because my situation is not terribly stressful, my mum’s okay. I’m just kind of managing it at a distance, and I’m retired, and I’ve got time, it’s had a small positive impact”.*

*(P4, daughter, aged 67)*


## 4. Discussion

Results from the first RCT evaluating clinical and cost effectiveness of the WHO’s iSupport programme showed no measurable improvements in carers’ level of depression and distress or in any of the secondary outcome measures. The current process evaluation findings help explain these outcomes and offer quality evidence to inform future research directions and development and implementation of iSupport, or similar web-based psychoeducational interventions.

The iSupport RCT was a pragmatic trial. Participants were given access to iSupport and, in line with its self-directed nature, had complete control over the time they chose to spend on it and how they chose to use it. Regardless of the availability of an ‘e-Coach’ (and reminder emails), usage data revealed that around half of intervention group participants spent a maximum of 1.5 h, and 18.3% of participants did not spend any time at all on iSupport. Our findings show that ‘available time’ was a vital factor influencing participants’ level of uptake. This highlights the importance of setting specific goals and having clear adherence expectations (e.g., minimum time commitment) as integral to the success of future research and implementation of self-directed e-health interventions such as iSupport. In other words, participants should receive clear information on what to expect from the intervention but also what minimum efforts will be required from them. This has been endorsed by other studies evaluating e-health interventions which have reported similar findings ([48]; [34]) including two reviews looking specifically at online interventions for dementia carers which advocate for the need to improve adherence monitoring and produce reliable data to explain it ([41]; [49]). The iSupport process evaluation aimed to explain and understand adherence, which has been defined as ‘*a vital and inseparable aspect of web-based interventions*’ ([27]). It used data from the web platform to investigate different aspects of how participants had engaged with iSupport (e.g., how many times participants logged on, what sections they accessed and completed, for how long, etc.). However, shortcomings of this data were identified, such as recording too much detail to allow consistent categorisation, which meant we were not able to produce a reliable picture of participants’ engagement with individual sections or modules. The disappointing quality of data is a familiar phenomenon for this type of self-directed, web-based interventions. Our findings add to the body of knowledge advocating for research efforts to improve the built-in auditing systems that e-health interventions should have ([27]; [28]; [37]).

### 4.1. Future Directions

As already discussed, RCT results concluded that iSupport is unlikely to be effective as a completely self-guided online intervention. Process evaluation findings support the need to improve iSupport for better usability and provide guidance that should be addressed in future adaptations. Firstly, complexity of navigation needs to be attended to so that participants can easily and straightforwardly choose and complete their preferred content and easily be able to ‘pick up from where they left it’. Secondly, the visual appearance of iSupport could be modified to both increase its appeal and improve functionality/layout across all types of devices that can be used to access it. Thirdly, the way in which iSupport provides feedback could be re-assessed. Participants did not consider that iSupport, in its current form, provided the right level of feedback, referring to standard (rather than personalised) responses following case-scenario tick-box exercises. iSupport does not provide any feedback following the completion of self-reflection activities. In order to address this, our participants advocated for a blended approach to delivery embedded within wider support systems. Similarly, participants who took part in a study on another web-based psychoeducational programme for unpaid carers of people with Alzheimer’s Disease ([14]) expected more dynamic content and further interaction with staff and peers. The most effective web-based intervention content and delivery method to support dementia carers has been identified in a recent umbrella review ([37]), which concluded that online interventions that are delivered online but include sessions with a personal element, such as telephone contact, showed the best results. [9] ([9]) showed a significant improvement in self-efficacy, mastery, and quality of life of carers of people with early-stage dementia after receiving the Partner in Balance programme which combines face-to-face coaching with tailored Web-based modules. Similarly, [7] ([7]), which examined the effectiveness of an online support intervention called Mastery over Dementia (MoD), concluded that it offered an effective treatment in reducing depression and anxiety in dementia carers, and their methodology included the guidance of a coach monitoring progress, providing feedback on homework, and sending reminders to participants. More recently evidence has identified clear benefits of a facilitator-enabled, online, multicomponent iSupport blended delivery of interventions to support dementia carers ([55]).

Although the UK iSupport RCT results showed no significant improvement in carers’ wellbeing measures compared with care as usual; the process evaluation findings suggest that the use of an online self-help psychoeducation tool such as iSupport can play a role in improving self-value, with carers feeling recognised and acknowledged. It could also potentially encourage carers to think about engaging in self-care activities (including help-seeking), which is associated with more positive perceptions of their caring role and less burden and depression ([29]). However, this process evaluation has highlighted the need to carefully consider carer characteristics when defining a target audience of an intervention. Participants in the iSupport RCT had already been carers for an average of three years, and as some reported during the qualitative interviews, they had already sought out support from different avenues, which meant that most of the information in iSupport was not new to them. From this, it can be suggested that iSupport may be more effective or relevant, leading to higher impact, if accessed in the early stages of the illness as the caring role develops. However, by assuming this, there could be a danger of ignoring the inherent complexity of interventions such as iSupport. Access to iSupport very soon after diagnosis could lead to a low level of uptake linked to difficulties accepting the diagnosis ([12]) and feelings of ‘too much information too soon’ as evidenced in our process evaluation. Our findings support research implementation efforts to explore timing factors in detail and in conjunction with a blended method of delivery, which could play a vital role in mitigating some of the risks just mentioned. For example, the involvement of a specialist nurse or other health professional could guide the carer on tailoring their use of iSupport to address their particular needs at that point in time. Although initial exploratory work is underway to evaluate this type of supported implementation of iSupport ([53]), more research is needed in this area.

### 4.2. Limitations

The iSupport RCT was a well-conducted, high-quality trial that recruited a large sample of dementia carers across the UK (n = 352). However, the sample lacked diversity, as participants were predominantly white, well-educated, and English-speaking ([51]). We acknowledge that future research efforts will need to consider additional engagement activities when designing strategies that enable successful recruitment of undeserved, diverse populations. These may include, for example, face-to-face visits to minority community and stakeholder groups and translation of advertising campaigns to a wide range of minority languages. Importantly, it is vital that research funders acknowledge the subsequent impact of these activities on budget and timelines.

The online survey that was used in this process evaluation included a validated tool, the SUS, but also non-validated, bespoke Likert scale and open-ended-type questions. We are aware that this can be seen as a limitation; however, as previously explained, the questions were designed to reflect the theoretical underpinnings of the CFIR and a detailed logic model which may counter some of the limitations.

All participants in the iSupport study received a £20 gift voucher for taking part regardless of which group they were randomised to or whether they actively engaged with iSupport. Hence, we are confident that it is very unlikely that this incentive had any impact on the results and level of use and acceptability of iSupport.

The process evaluation that ran alongside the iSupport trial was methodologically sound and reached all data collection targets, hence generating high quality reliable evidence. We acknowledge that data generated during the process evaluation might not have reflected the experience of all participants. However, we are confident that the purposive sampling and the mixed-methods approach adopted ensured representation of different carer experiences.

## 5. Conclusions

In light of the RCT trial results, this process evaluation has proven invaluable at providing explanatory pathways on why iSupport, which was expected to produce positive health impacts, did not do so. The iSupport intervention was well received by a majority of dementia carers who took part in the trial. However, synthesis across all process evaluation data sources suggests that e-health interventions such as iSupport which are 100% self-guided may not generate significant outcomes in the complex context of dementia care unless steps are taken to improve their uptake and usability. Given the current global implementation of iSupport by the WHO, and considering the findings of this process evaluation, efforts need to focus on exploring delivery and implementation pathways that include self-directed approaches as well as other interactive components, such as contact with a health professional (or similar). This professional will, for example, provide guidance on the use and tailoring of iSupport to the particular needs of that carer as well as feedback following completion of iSupport exercises. If blended approaches are successfully implemented in iSupport, further evaluation of the effectiveness of the intervention will be warranted. This process evaluation has provided findings to assist the interpretation of the iSupport RCT results, to inform the direction of the global implementation of iSupport by the WHO, and to suggest prioritisation of resources and effective interventions by national health and care providers.

## Figures and Tables

**Figure 1 behavsci-15-01107-f001:**
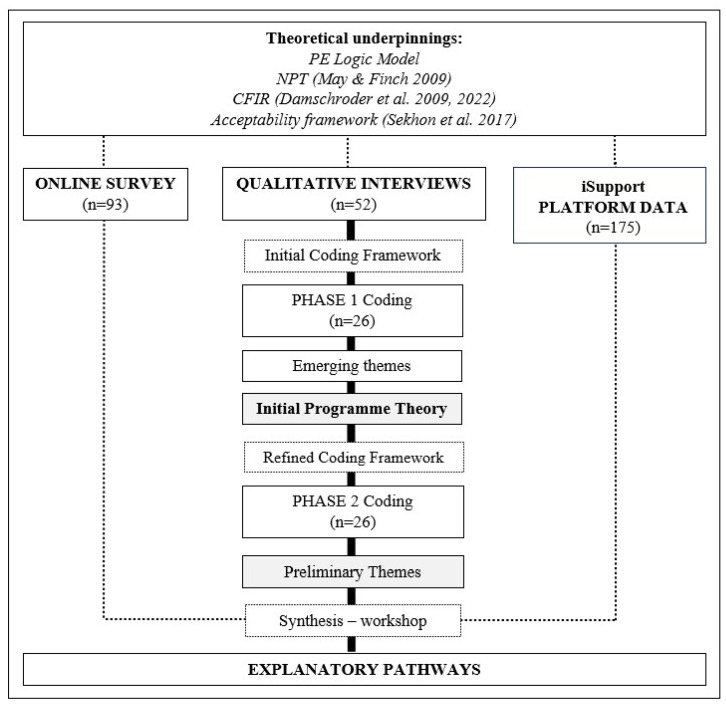
Mixed-method design and theoretical underpinnings ([32]; [45]; [15], [16]) of the iSupport process evaluation (PE: Process Evaluation; CFIR: Consolidated Framework of Implementation Research; NPT: Normalization Process Theory).

**Figure 2 behavsci-15-01107-f002:**
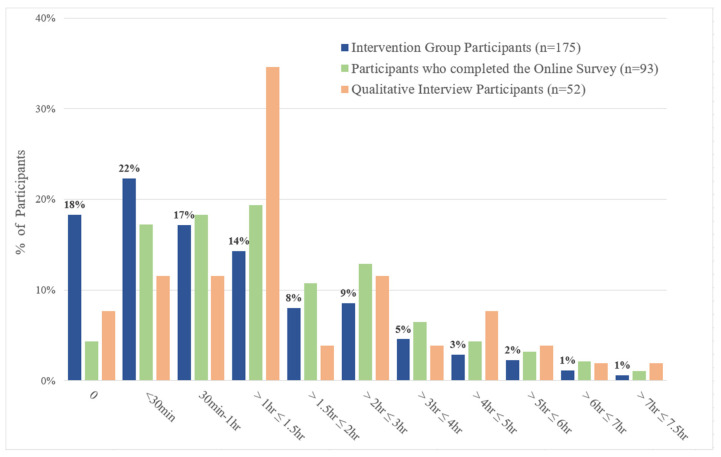
Total amount of time that participants spent on iSupport (iSupport usage data) divided in three groups: participants in the intervention group, participants who completed the Online Survey, and participants who took part in a qualitative interview.

**Figure 3 behavsci-15-01107-f003:**
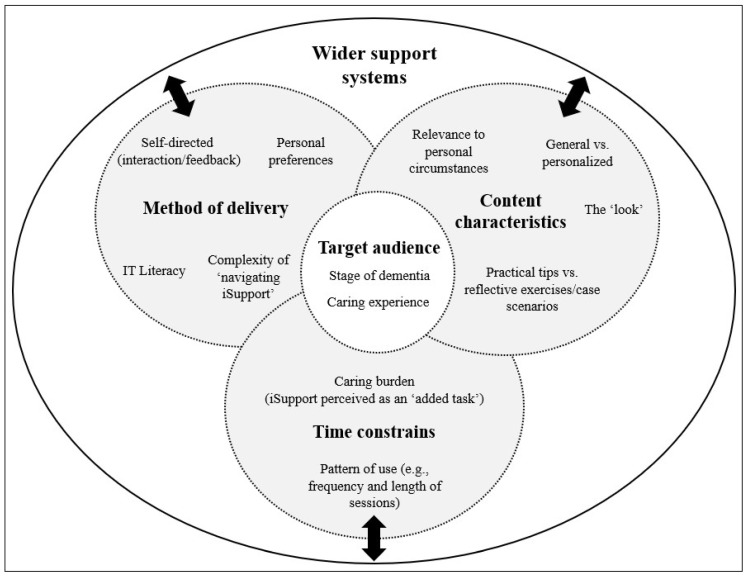
Factors driving the uptake and level of engagement with iSupport by study participants.

**Table 1 behavsci-15-01107-t001:** Process evaluation: participants’ demographic characteristics and baseline data (SD: standard deviation; IQR: interquartile range: PLWD: person living with dementia; NVQ: national vocational qualification).

	Interview Part. (N = 52)	Online Survey Part. (N = 93)
	Mean (SD)/Median [IQR]	Mean (SD)/Median [IQR]
Age (years)	62.6 (10.5)/62 [55, 72]	62.8 (10.6)/62 [56, 71]
ZBI-12	20.8 (8.7)/21 [16, 26]	20.8 (8.3)/22 [15, 26]
CESD-10	10.5 (6.7)/9 [6, 15]	10.3 (6.3)/10 [6, 14]
Years caring	4.1 (3.9)/3 [1.5, 5]	4.1 (4)/2.5 [1.2, 5]
	N (%)	N (%)
Gender		
Female	36 (69.2)	75 (80.6)
Male	16 (30.8)	18 (19.4)
Relationship to PLWD		
Spouse/partner	24 (46.2)	40 (43)
Child	27 (51.9)	48 (51.6)
Other	1 (1.9)	5 (5.4)
Ethnic group		
White British	47 (90.4)	89 (95.7)
White Irish	1 (1.9)	1 (1.1)
Any other White background	1 (1.9)	1 (1.1)
Indian	1 (1.9)	1 (1.1)
Any other Asian background	1 (1.9)	0 (0)
Any other ethnic group	1 (1.9)	1 (1.1)
Level of education		
University Higher Degree (MA; MSc; PhD)	16 (30.8)	22 (23.7)
First degree level qualification (BA; BSc)	14 (26.9)	35 (37.6)
Apprenticeship	1 (1.9)	1 (1.1)
Degree level (NVQ Level 4; teaching/nursing)	8 (15.4)	12 (12.9)
AS, A Level, Baccalaureate	4 (7.7)	7 (7.5)
NVQ level 3 or below	2 (3.8)	3 (3.2)
Any other qualification	7 (13.5)	13 (14)
Employment		
Not in paid work	2 (3.8)	4 (4.3)
Not in paid work (retired)	31 (59.6)	56 (60.2)
Not in paid work (seeking work)	1 (1.9)	1 (1.1)
Paid work (full-time)	9 (17.3)	14 (15.1)
Paid work (part-time)	9 (17.3)	18 (19.4)

**Table 2 behavsci-15-01107-t002:** Online survey responses (non-SUS items) of those participants who spent a minimum of 30 min on iSupport (n = 73).

How Relevant Are the iSupport Modules to Your Role as a Carer?	Not Relevant	2	3	4	Extremely Relevant	No Answer
	n (%)	n (%)	n (%)	n (%)	n (%)	n (%)
Module 1: Introduction to dementia	5 (6.8)	6 (8.2)	16 (21.9)	13 (17.8)	31 (42.5)	2 (2.7)
Module 2: Being a carer	2 (2.7)	7 (9.6)	14 (19.2)	16 (21.9)	31 (42.5)	3 (4.1)
Module 3: Caring for me	4 (5.5)	4 (5.5)	20 (27.4)	13 (17.8)	29 (39.7)	3 (4.1)
Module 4: Providing everyday care	4 (5.5)	6 (8.2)	14 (19.2)	11 (15.1)	34 (46.6)	4 (5.5)
Module 5: Dealing with behaviour changes	4 (5.5)	4 (5.5)	15 (20.5)	10 (13.7)	34 (46.6)	6 (8.2)

**Table 3 behavsci-15-01107-t003:** Online survey responses (non-SUS items) of those participants who spent a minimum of 30 min on iSupport (n = 73).

	Not at All	2	3	4	Very Much
	n (%)	n (%)	n (%)	n (%)	n (%)
Anxiety	36 (49.3)	15 (20.5)	12 (16.4)	9 (12.3)	1 (1.4)
Depression	39 (53.4)	17 (23.3)	12 (16.4)	5 (6.8)	0 (0)
Stress levels	33 (45.2)	19 (26)	12 (16.4)	8 (11)	1 (1.4)
Feelings of burden	34 (46.6)	15 (20.5)	13 (17.8)	11 (15.1)	0 (0)
Sleep	41 (56.2)	16 (21.9)	13 (17.8)	3 (4.1)	0 (0)
Relationship with the person that you care for	21 (28.8)	10 (13.7)	17 (23.3)	19 (26)	6 (8.2)
Your confidence as a carer	17 (23.3)	9 (12.3)	14 (19.2)	23 (31.5)	10 (13.7)

## Data Availability

The data sets generated and analysed during the study are available upon request. All data requests should be submitted to the corresponding author for consideration. Access to anonymised data may be granted following review.

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
