# Peer review of "A Process Evaluation of the UK Randomised Trial Evaluating ‘iSupport’, an Online e-Health Intervention for Adult Carers of People Living with Dementia"

_behavsci, 2025, doi:10.3390/bs15081107_

Round 1

Reviewer 1 Report

Comments and Suggestions for Authors

The manuscript describes the process evaluation of the iSupport intervention, using mixed-methods. It is a well written and reflective manuscript. I have a few minor comments and suggestions that would be useful to address:

Results, Line 234-235: The authors report a difference between younger and older participants on the SUS. Raising this point felt slightly ad-hoc and perhaps not systematic (did the authors look at other factors that might differentiate the SUS scores?). It also raised the question about whether the authors could run a non-parametric statistical comparison between these groups.  

Results: Would it be possible to explore whether SUS scores were associated with a) length of time using iSupport, and b) whether they had to contact the e-Coach or not. This might help better understand if engagement was linked with perceived useability.

Results: Please provide basic demographic information alongside participant IDs when providing quotes, as they can help provide context and perhaps flag patterns (e.g. P1, daughter, aged 45)

Results, Line 412: I believe the frequencies reported in the parenthesis is missing “n=”

Figure 2. Please provide a bit more of a description of what the figure shows than “iSupport usage data” so that it is more easily interpreted stand alone.

Discussion: The limitations section could be developed further to better reflect on limitations, One example being the adoption of some non-validated quantitative outcome measures.

Author Response

Dear reviewer, thank you very much for your constructive comments and feedback which we believe have increased the quality of the manuscript significantly.

We have tried to reply to all of your comments with as much detail as possible. Please find our responses below and as ‘tracked changes’ in the manuscript. 

COMMENT 1: Results, Line 234-235: The authors report a difference between younger and older participants on the SUS. Raising this point felt slightly ad-hoc and perhaps not systematic (did the authors look at other factors that might differentiate the SUS scores?). It also raised the question about whether the authors could run a non-parametric statistical comparison between these groups.  

RESPONSE 1: Thank you very much for raising this issue. When analysing the data from the SUS scores we worked with data from the 73 participants who had spent 30 minutes or more on iSupport. Of these, n=16 males and n=57 females and n=42 18-64 yr and n=31 65+ yr. We have completed non-parametric Spearman’s correlations tests and found no correlation between SUS scores and age or time spent on iSupport. The manuscript has been edited to include this information (Line 249)

COMMENT 2: Results: Would it be possible to explore whether SUS scores were associated with a) length of time using iSupport, and b) whether they had to contact the e-Coach or not. This might help better understand if engagement was linked with perceived useability.

RESPONSE 2: Due to the sample characteristics we were not able to run any tests to confirm whether contacting the e-Coach was associated with the SUS score (of the 73 participants 33 contacted the e-Coach). The manuscript has been edited to include the lack of correlation between time on iSupport and SUS scores and detail on the use of the e-Coach (Line 245-249)

COMMENT 3: Results: Please provide basic demographic information alongside participant IDs when providing quotes, as they can help provide context and perhaps flag patterns (e.g. P1, daughter, aged 45)

RESPONSE 3: The manuscript has been edited accordingly, demographic information (gender, relationship with person living with dementia and age) has been added alongside the participant IDs. Changes start on first quote in Line 287

COMMENT 4: Results, Line 412: I believe the frequencies reported in the parenthesis is missing “n=”

RESPONSE 4: Thank you, the manuscript has been edited accordingly (Line 438)

COMMENT 5: Figure 2. Please provide a bit more of a description of what the figure shows than “iSupport usage data” so that it is more easily interpreted stand alone.

RESPONSE 5: Thank you very much, a longer description has been added (Line 270)

COMMENT 6: Discussion: The limitations section could be developed further to better reflect on limitations, One example being the adoption of some non-validated quantitative outcome measures.

RESPONSE 6: Thank you very much, we have added more detail accordingly (Line 613)

Reviewer 2 Report

Comments and Suggestions for Authors

Thank you for the opportunity to review this manuscript, and congratulations to the authors on their work. The manuscript presents the process evaluation of a randomized controlled trial (RCT) on iSupport, a web-based platform originally developed by the World Health Organization (WHO) to support informal caregivers of people living with dementia. I consider this study highly relevant, as iSupport has gained worldwide attention and has been adapted for use in over 40 countries. The authors have conducted a high-quality study using a robust mixed-methods design. The results are highly relevant, as they address key concerns regarding the impact of iSupport on caregivers’ well-being and highlight important considerations for future research and decision-making—not only concerning the implementation of iSupport, but also for other web-based psychoeducational interventions.

No major concerns were identified, and I found the manuscript to be well written, with a strong methodological foundation. Nonetheless, I would like the authors to clarify two minor points:

  1. Regarding the following paragraph:

“During Phase 1, two researchers (PM-A and MC) coded half of the transcripts independently (n=26) and assigned relevant data extracts to each code. The distribution of codes was recorded, and new codes were created for data falling outside the coding framework to avoid missing important concepts. Emerging themes were identified as meaningful patterns across coded data and as a result an initial programme theory and a refined coding framework were developed. Phase 2 involved the thematic analysis of the remaining transcripts (n=26) applying the refined coding framework. An iterative process of review and discussion among process evaluation research team members was followed to agree a set of preliminary themes (Supplementary File S5)”

It is unclear why the thematic analysis was conducted in two distinct phases. If the intention was to use the initial coding to refine the framework by identifying emerging codes, could you clarify whether the first 26 interviews were reanalysed using the final coding framework? Inconsistencies in coding across phases could result in differences in findings, and it would be helpful to understand how this potential issue was addressed.

  1. In the results, time constraints were identified as a barrier to engaging with iSupport. According to your findings, iSupport was “described as an ‘added task’ to an already busy schedule.” The following two quotes were provided to support this theme:

 “To me it became another chore if you like. It became another task in a day that I needed 266 to do. And it was a task that actually I could say well, I haven’t got time to do that, or 267 that’s not the best use of my time today”. P1 268

“I am trying to juggle everything, and do this, and it’s nice to have the support there, for 269 someone to say you have to value yourself, and you can have time out to do things, but 270 in reality, it just feels like another pressure, and it really gets on my nerves.” P2

I wonder whether participants received any benefit or compensation for their participation in the study, as this may have influenced their perception of iSupport as an additional burden or task. It is possible that participants felt obligated or pressured to complete the intervention for this reason. If this was the case, it would be appropriate to acknowledge this in the study’s limitations.

Author Response

Dear reviewer, thank you very much for your constructive comments and feedback which we believe have increased the quality of the manuscript significantly.

We have tried to reply to all of your comments with as much detail as possible. Please find our responses below and as ‘tracked changes’ in the manuscript. 

COMMENT 1:

Regarding the following paragraph:

“During Phase 1, two researchers (PM-A and MC) coded half of the transcripts independently (n=26) and assigned relevant data extracts to each code. The distribution of codes was recorded, and new codes were created for data falling outside the coding framework to avoid missing important concepts. Emerging themes were identified as meaningful patterns across coded data and as a result an initial programme theory and a refined coding framework were developed. Phase 2 involved the thematic analysis of the remaining transcripts (n=26) applying the refined coding framework. An iterative process of review and discussion among process evaluation research team members was followed to agree a set of preliminary themes (Supplementary File S5)”

It is unclear why the thematic analysis was conducted in two distinct phases. If the intention was to use the initial coding to refine the framework by identifying emerging codes, could you clarify whether the first 26 interviews were reanalysed using the final coding framework? Inconsistencies in coding across phases could result in differences in findings, and it would be helpful to understand how this potential issue was addressed.

RESPONSE 1: Thank you very much for your comment. We have edited that paragraph to make the process clearer (Line 193). The thematic analysis was indeed carried out in two phases which allowed for tailoring and for the addition of codes that were missing in the initial coding framework as the first 26 interviews were completed. During Phase 2 the remaining 26 interviews were then coded using the refined coding framework which was very similar to the initial one with some added/refined codes. We are confident that there were no inconsistencies of coding across the two phases as the codes that were added/refined after Phase 1 were minimal and no codes were removed.  We are confident that the steps we followed provided us with a rigorous analysis that led to in-depth refined themes (Supplementary file S5).

COMMENT 2: In the results, time constraints were identified as a barrier to engaging with iSupport. According to your findings, iSupport was “described as an ‘added task’ to an already busy schedule.” The following two quotes were provided to support this theme:

 “To me it became another chore if you like. It became another task in a day that I needed 266 to do. And it was a task that actually I could say well, I haven’t got time to do that, or 267 that’s not the best use of my time today”. P1 268

“I am trying to juggle everything, and do this, and it’s nice to have the support there, for 269 someone to say you have to value yourself, and you can have time out to do things, but 270 in reality, it just feels like another pressure, and it really gets on my nerves.” P2

I wonder whether participants received any benefit or compensation for their participation in the study, as this may have influenced their perception of iSupport as an additional burden or task. It is possible that participants felt obligated or pressured to complete the intervention for this reason. If this was the case, it would be appropriate to acknowledge this in the study’s limitations.

RESPONSE 2: Thank you for raising an important point.  All participants that took part in the iSupport RCT trial received a £20 gift voucher for taking part in the study (all details in the main paper Windle et al. The Lancet Regional Health 2025 - https://www.thelancet.com/journals/lanepe/article/PIIS2666-7762(24)00293-X/fulltext). We see this as a very small incentive that everyone received (whether or not they had actually logged on to use iSupport or not) hence we can be quite confident that there is very little chance that this made an impact on how participants perceived iSupport as a burden or a task that they felt forced to complete. We have acknowledged this in the limitations section (Line 618).